# A Succinct Method for Non-Line-of-Sight Mitigation for Ultra-Wideband Indoor Positioning System

**DOI:** 10.3390/s22218247

**Published:** 2022-10-27

**Authors:** Ang Liu, Shiwei Lin, Jianguo Wang, Xiaoying Kong

**Affiliations:** 1Faculty of Engineering and Information Technology, University of Technology Sydney, 81 Broadway, Ultimo, Sydney, NSW 2007, Australia; 2School of IT and Engineering, Melbourne Institute of Technology, Sydney Campus, Sydney, NSW 2007, Australia

**Keywords:** UWB, NLOS, delay model, WLS

## Abstract

Ultra-wideband (UWB) is a promising indoor position technology with centimetre-level positioning accuracy in line-of-sight (LOS) situations. However, walls and other obstacles are common in an indoor environment, which can introduce non-line-of-sight (NLOS) and deteriorate UWB positioning accuracy to the meter level. This paper proposed a succinct method to identify NLOS induced by walls and mitigate the error for improved UWB positioning with NLOS. First, NLOS is detected by a sliding window method, which can identify approximately 90% of NLOS cases in a harsh indoor environment. Then, a delay model is designed to mitigate the error of the UWB signal propagating through a wall. Finally, all the distance measurements, including LOS and NLOS, are used to calculate the mobile UWB tag position with ordinary least squares (OLS) or weighted least squares (WLS). Experiment results show that with correct NLOS indentation and delay model, the proposed method can achieve positioning accuracy in NLOS environments close to the level of LOS. Compared with OLS, WLS can further optimise the positioning results. Correct NLOS indentation, accurate delay model and proper weights in the WLS are the keys to accurate UWB positioning in NLOS environments.

## 1. Introduction

In recent decades, the Internet of Things (IoT) and intelligence industry developed rapidly. Indoor position systems (IPSs) are essential technological infrastructures for them. Both industry and academia are improving the accuracy of IPSs by using different technologies and algorithms to expand their application scenarios [1]. The Global Navigation Satellite System (GNSS) can provide positions, time service and navigation in the outdoor environment. In the indoor area, buildings block the satellite signal leading to poor accuracy, so GNSS is not a good choice for IPSs [2]. The mainstream of indoor technology mainly classifies two categories; the first one uses the principle that sound and light will reflect when encountering obstacles, with the representation of ultrasound and optical systems. The range information can be derived to resolve the target coordinates by measuring the fly time of the sound or light. Although these systems can be applied to some unstructured and uncertain environments depending on their mapping ability, a serious drawback is that ultrasound and light cannot penetrate the obstacles, which limits those techniques to apply to IPSs [3]. The second category uses the radio signal to locate the target, such as WiFi [4,5], Bluetooth [6,7] and Ultra-wideband (UWB). Many previous research works explained the advantages and disadvantages of these methods in detail. Applying WiFi to IPSs has the following benefits: wide distribution, low hardware cost, and high flexibility, but there are still some challenges like multipath, low accuracy and device heterogeneity [4]. The main advantages of Bluetooth technology are low hardware cost, low energy consumption, small size and high security [6,7,8]. On the other side, it still has some unavoidable drawbacks, like the calculation of the fingerprint algorithm increases dramatically when placed in a large indoor area. Furthermore, it is hard to improve the position accuracy to a centimetre lever [7]. UWB technology is up-and-coming for IPSs compared with those technologies.

UWB is a feasible technology for localisation and target tracking on account of properties. For instance, the UWB uses an extensive baseband to communicate between tags and anchors. The baseband pulse duration is nanoseconds, so the UWB has a high time resolution to tackle the problem of multipath and through wall propagation [9]. UWB has the same virtues of low power consumption and hardware costs like Bluetooth and WiFi but more accuracy.

Time of Arrival (TOA) [10,11], Time Difference of Arrive (TDOA) [12], Angle of Arrival (AOA) [12], and Two Way Range (TWR) [13] are the commonly used algorithms for UWB IPSs. The range measurements are derived from the signal’s time of flight (TOF) between fixed anchors and mobile tags. So, any factors that affect the TOF measurement will impact the accuracy of the measured distance and the position, such as multipath, non-line-of-sight (NLOS), rugged to synchronise between tags and anchors, and interference by other radio waves [14]. The range error caused by NLOS appears when the direct path of the signal between tags and anchors is blocked, and the measured TOF includes extra time delay. The range error of NLOS is always an enlargement and can reach a few meters, which is the most significant factor harming the accuracy of UWB IPSs. In addition, NLOS is unavoidable in real scenarios, so the reduction of NLOS error is of great importance.

Improving the performance of UWB IPSs in the NLOS environment can be divided into two steps, identification and mitigation. Many approaches have been proposed to identify the NLOS and mitigate its error, which can be grouped into three categories. The first category is to analyse the statistical channel characteristics [15] using machine learning strategies like deep learning [16], Support Vector Machine (SVM) [17] and Import Vector Machine (IVM) [18]. The second category is using external constraints to detect the NLOS and then using the range error model to mitigate it. In the literature [2], the NLOS can be recognised by the different relative heading angle (RHA) as the threshold; the NLOS range error is compensated by experimental methodology. The last category draws support from other sensors and maps. Many auxiliary sensors like IMU [19], V-SLAM [20] and Lidar SLAM [21] have been proposed to cope with the decline in location accuracy of UWB IPSs in NLOS scenarios.

All those methodologies above have some inherent vice. For the first category, simulation results in the [16] show the accuracy of identifying NLOS by different machine learning methods: 86.94% for convolutional neural networks (CNN), 78.51% for stacked long short-term memory (stacked-LSTM) and 84% for combined CNN and LSTM. Some machine learning methods claim can reach 90% accuracy, which positively correlates with the amount of the database. To get the above result in [16], analysing 126,000 channel impulse responses cost CNN 21.84 s, stacked-LSTM over 200 s and CNN + LSTM over 15 s. Due to a large amount of data and the high time cost, those methods are hard to apply to real-time locations that need fast reaction time. The second category has a better response time but still has some limitations. The experiment results in [22] show that over 90% of NLOS cases can be recognised by the Fresnel zone and threshold from prior knowledge. Still, the Fresnel zone has strict restrictions on the application that cannot be widely used in real scenarios. A consensus is that multisensor fusion can improve IPSs’ accuracy, availability and reliability [23], which will be the future research plan for this project.

This paper proposes a novel, succinct method to identify and mitigate the NLOS error in real time. The identification step is based on the principle that the measured distance changes rapidly when a switch between NLOS and LOS happens for a moving tag. The variance of the difference of ranges observed from the same anchor between two adjacent time slots is relatively constant in LOS, and it will bump up when a range contains NLOS error. By this means, NLOS can be distinguished. For NLOS error mitigation, we propose a delay model to correct the NLOS range error caused by a wall, which includes the factors of incident angle, wall thickness and material to calculate the delay. The delay model is derived from a significant amount of experimental data. The corrected range data by this model is then used to compute the coordinate of the mobile tag, using ordinary least square (OLS) and weight least square (WLS). Figure 1 is the flowchart of the proposed method.

This flowchart illustrates the strategy to deal with NLOS proposed in this paper. The first step is setting a threshold from prior experience to estimate whether NLOS occurs. If no measurement is affected by NLOS, the tag’s position can be computed directly using the OLS range measurements. If UWB IPSs have more than three fixed anchors, even if NLOS happens for some anchors, the OLS algorithm can still accurately position the tag when at least three anchors are in LOS. If the system cannot receive enough LOS range measurements, the data containing NLOS will be processed by the delay estimation model. The delay model will correct those NOLS errors and estimate the uncertainty of the corrected ranges. Then, WLS is applied to calculate the UWB tags’ position.

The rest of this paper is organised as follows. Section 2 presents the algorithms used in this paper, including the sliding window method to identify NLOS, the delay model and position algorithms. Section 3 gives the experiment design and results, which verify to what degree the delay model and WLS can improve the position accuracy. Section 4 discusses the limitation of IPSs with only UWB and the future research plan. The last part is the conclusion.

## 2. Algorithms and Modeling

### 2.1. Position Algorithms

The position algorithms used in this paper are OLS for at least three anchors in LOS environments and WLS for less than three anchors in LOS environments. The following figure illustrates the geometric position of fixed anchors and tags.

Figure 2 illustrates the geometric relationship between fixed anchors (A, B and C) and a tag (D). The coordinate of anchor *n* is $\left(x_{n},y_{n}\right)\;n=1,2,3\ldots$, the coordinate of the tag is (*x*, *y*), and $d_{n}$ is the distance between tags and anchors, which can be computed by Equation (1).
(1)$$d_{n}=t_{n}*C$$
where $t_{n}$ is TOF between tag and anchor n, and C is the speed of light.

#### 2.1.1. OLS

OLS is the most common approach for mathematical optimisation. The core of this algorithm is minimising the residual to realise the optimal solution. Equation (2) is the mathematical form for Figure 2.
(2)$$\{\begin{array}{c} {\left(x-x_{1}\right)}^{2}+{\left(y-y_{1}\right)}^{2}=d_{1}^{2}\\ {\left(x-x_{2}\right)}^{2}+{\left(y-y_{2}\right)}^{2}=d_{2}^{2}\\ \cdots \\ {\left(x-x_{n}\right)}^{2}+{\left(y-y_{n}\right)}^{2}=d_{n}^{2} \end{array}$$

Equation (2) can draw from the geometric relationship between anchors and tags and expand Equation (2) to get Equation (3).
(3)$$\{\begin{array}{c} 2(x_{2}-x_{1})x+2(y_{2}-y_{1})y=d_{1}^{2}-d_{2}^{2}-(x_{1}^{2}+y_{1}^{2})+(x_{2}^{2}+y_{2}^{2})\\ 2(x_{3}-x_{2})x+2(y_{3}-y_{2})y=d_{2}^{2}-d_{3}^{2}-(x_{2}^{2}+y_{2}^{2})+(x_{3}^{2}+y_{3}^{2})\\ \cdots \\ 2(x_{n}-x_{n-1})x+2(y_{n}-y_{n-1})y=d_{n-1}^{2}-d_{n}^{2}-(x_{n-1}^{2}+y_{n-1}^{2})+(x_{n}^{2}+y_{n}^{2}) \end{array}$$

Use the matrix to represent Equation (3) as follows:
(4)AX=B
(5)$$r_{n}=x_{n}^{2}+y_{n}^{2}$$
(6)$$r=x^{2}+y^{2}$$
(7)$${\left(x-x_{n}\right)}^{2}+{\left(y-y_{n}\right)}^{2}=d_{n}^{2}$$
(8)$$-2x_{n}x-2y_{n}y+r=d_{n}^{2}-r_{n}$$
(9)$$A=\left[\begin{array}{ccc} -2x_{1} & -2y_{n} & 1\\ \ldots & \ldots & \ldots \\ -2x_{n} & -2y_{n} & 1 \end{array}\right]$$
(10)$$X=[\begin{array}{c} x\\ y\\ r \end{array}]$$
(11)$$B=\left[\begin{array}{c} d_{1}^{2}-r_{1}\\ \ldots \\ d_{n}^{2}-r_{n} \end{array}\right]$$

In practical application, those three circles have a great chance will not intersect at a point due to the measurement noise [24]. When considering the effect of noise (*e*), Equation (4) should present as follows:(12)e= AX−B

When the partial derivative of squared error is 0, the error value is the smallest.
(13)$$\frac{\partial e^{2}}{\partial X}=2A^{T}AX-2A^{T}B=0$$


From the previous equation can calculate the tag coordinate *X*:(14)$$X={\left(A^{T}A\right)}^{-1}\left(A^{T}B\right)$$

The OLS algorithm treats all measurement values equally and can perform well in LOS environments due to the nearly identical noise level. The measurement noise in NLOS environments will interfere with the range value. Thus the OLS may not perform well in NLOS. Contrapose that phenomenon, the WLS algorithm may be more suitable for UWB IPSs in NLOS environments.

#### 2.1.2. WLS

This paper use WLS to calculate the tag’s coordinate for further improvement in the NLOS scene. In the NLOS environment, the range value may contain a colossal measurement error that cannot be directly used for position. Giving corresponding weight to different anchors’ range values can improve the position accuracy. The method of determining the weights for each measurement will be explained in detail in Section 3.4.

The weight matrix for the measured ranges from anchors can represent by Equation (15).
(15)$$W=diag\left[W_{1},W_{2},W_{3}\ldots W_{n}\right]$$

Based on the OLS, the partial derivative equation of WLS is:
(16)$$\frac{\partial e^{2}}{\partial X}W=2A^{T}WAX-2A^{T}WB=0$$

The following equation calculates the tag’s coordinates:
(17)$$X={\left(A^{T}WA\right)}^{-1}\left(A^{T}WB\right)$$

The premise of applying the WLS algorithm is to identify the NLOS accurately because the weights for range values in LOS and NLOS are significant differences.

### 2.2. Proposed Method to Identify NLOS

The proposed method to identify NLOS is based on the variance of the difference in measured distance between two adjacent sample times. The measured distance between anchor and tag at the time n can be modelled as $\hat{d_{n}}$ which equal to actual distance ($d_{n}$) plus measurement noise (*ε*) in the LOS environment. The measured distance in NLOS ($\hat{d_{n}}$) equal to actual distance ($d_{n}$) plus measurement noise (*ε*) superadd delay caused by NLOS ($\varepsilon_{NLOS}$). So, the difference in measured distance between two adjacent time slots can be computed as:


(18)
$$\Delta \hat{d_{n}}=|\hat{d_{n}}-{\hat{d}}_{n-1}|=\{\begin{array}{cc} d_{n}-d_{n-1} & LOS(n-1)\to LOS(n)\\ d_{n}-d_{n-1}+\varepsilon_{NLOS} & LOS(n-1)\to NLOS(n) \end{array}$$


The UWB module used in this paper is DecaWave DW1000 which is compliant with IEEE 802.15.4-2011 UWB standard [14]. The sampling frequency is 3 Hz, moreover most indoor robot speed is less than 1 m/s, so $d_{n}-d_{n-1}$ should be a tiny number around a few centimetres. But $\varepsilon_{NLOS}$ is around decimetres to a few meters, which is much greater than $d_{n}-d_{n-1}$. The method of the sliding window of variance of $\Delta \hat{d_{n}}$
was proposed in this paper to identify NLOS, which is illustrated in Figure 3. The threshold used to estimate the NLOS is the variance of $\Delta \hat{d_{n}}$ measured in a LOS environment (Var($\Delta \hat{d_{n}}$ (LOS))).

For further annotation of this method in Figure 3, when a tag was not in NLOS until time slot n, the Var($\Delta \hat{d_{n}}$) will suddenly increase due to the measured distance $d_{n}$ was added a huge NLOS error. Indeed, the value of Var($\Delta \hat{d_{n}}$) will be significantly greater than the threshold. Detecting the Var($\Delta \hat{d_{n}}$) is bigger than the threshold means this measured data would most probably affect by NLOS. Due to the great measurement noise in NLOS, the measurement range will keep jittering in an extensive range leading to the value of Var($\Delta \hat{d}$) still greater than the threshold in this situation. If tag back to LOS from NLOS at time slot k, the value of Var($\Delta {\hat{d}}_{k+3}$) will have a significant drop to be smaller than the threshold. The experiment will verify this strategy to identify the NLOS.

### 2.3. Delay Model for NLOS Mitigation

This paper mainly focuses on the NLOS caused by walls. The strategy to compensate for the NLOS delay is deriving a delay model from measured distance in the NLOS condition. The following figure illustrates the geometric model.

Figure 4 and Figure 5 show a universal model for all sorts of NLOS caused by a wall. In this model, the wall has an angle $\theta_{3}$ between the positive *x*-axis, anchors and tags are at different heights, where *A* and *T* represent the anchor point and tag point, respectively. Line $TT^{\prime }$ perpendicular to the wall and $AT^{\prime }$ perpendicular to $TT^{\prime }$.
$A^{\prime }$ is the projection point of A on plan y=tz. $\theta_{1}$ is the angle between line AT and $AT^{\prime }$; $\theta_{2}$ is the angle between line $AT^{\prime }$ and positive y-axis; $\theta_{4}$ is the angle between the normal of the wall and $A^{\prime }T$; $\theta_{5}$ is the angle between the normal of the wall and AT. $d_{wall}$ is the thickness of the wall, and $d_{NLOS}$ is the actual distance of UWB signal propagation in the wall.

The primary factors leading to NLOS errors are the electrical permittivities and the thickness of a material when the UWB signal propagates through this obstacle. The electrical permittivity for a wall is a constant number in this experiment, and the relative thickness of the wall is increased with the incident angle increasing when the UWB signal penetrates the wall. The NLOS delay model is defined in Equation (19)
(19)$$\Delta d_{NLOS}=A*d_{NLOS}*\left(\sqrt{\epsilon_{wall}}-1\right)+B$$
where $\Delta d_{NLOS}$ is the NLOS delay caused by a wall, $\epsilon_{wall}$ is the permittivity of the wall.
(20)$$d_{NLOS}=\frac{d_{wall}}{cos\theta_{4}*cos\theta_{1}}=\frac{d_{wall}}{cos\theta_{5}}$$
(21)$$\theta_{1}=\arcsin\left(\frac{\left|az-tz\right|}{\sqrt{{\left(ax-tx\right)}^{2}+{\left(ay-ty\right)}^{2}+{\left(az-tz\right)}^{2}}}\right)$$
(22)$$\theta_{2}=\arctan\left(\frac{\left|ax-tx\right|}{\left|ay-ty\right|}\right)$$


The relationship between $\theta_{4}$, $\theta_{2}$ and $\theta_{3}$ showing in Equation (23)
(23)$$\theta_{4}=\{\begin{array}{c} \theta_{2}-\theta_{3}\;\;\;\;if\;\theta_{2}>\theta_{4}\\ \theta_{2}+\theta_{3}\;\;\;\;if\;\theta_{2}\leq \theta_{4} \end{array}$$

The delay model is defined as follows:(24)$$\Delta d_{NLOS}=A*\frac{d_{wall}*\left(\sqrt{\epsilon_{wall}}-1\right)}{cos\theta_{5}}+B$$


The measured data were used to ascertain the coefficients A and B in Equation (24), a tag was placed in 45 different positions with various incident angles covering the range from 0° to 80°, and the LOS path was blocked by different wall thicknesses (0.16 m and 0.26 m) with permittivity setting to 5.5. At least 60 samples were taken at each position. By computing the fitting equation from those more than 2700 data, the coefficient A equals 0.3459, and B equals 0.2722. Consequently, the corrected distance can calculate by the NLOS delay subtracted from the measurement ranges. As the transmission distance of UWB can reach several tens of meters, this model can be applied to different indoor scenarios, such as smart homes and intelligent warehouses. The model can be applied to homogeneous media, but the coefficients are different for different environments, so it is necessary to collect data to calculate the coefficients before applying the model. This is also an unavoidable disadvantage of the model, as the calculated delays will be inaccurate when the medium is not homogeneous in certain areas. It is difficult to solve all problems with UWB alone, and the best approach is to fuse multiple sensors, which will be discussed in future research.

## 3. Experiment and Results

### 3.1. Experimental Environments and Equipment

This experiment is conducted in an apartment (Figure 6e) with the following dimensions: 7.8 m wide and 5.848 m long, as shown in Figure 6a,b. Four fixed anchors (Figure 6c–e) amount in different positions, represented by the yellow square in Figure 6a,b. The coordinates of Anchors 0, 1, 2 and 3 for experiment 1 are (1.19, 5.75, 1.5), (0.41, 3.95, 1.5), (4.27, 4.95, 1.5), (5.41, 1.75, 1.5), respectively. In experiment 2, the coordinate of Anchor 1 changes to (0.49, 2.65, 1.5), leading to only one LOS anchor in Area 3. These two experiments cover all conditions of NLOS anchors’ number in UWB IPSs, shown in Table 1.

The purple line in Figure 6a,b are the reference paths, which start from Area 2 to Area 3 through Area 1 and then go back to the endpoint at Area 5 through Area 1 and 4 when reaching the stop point in Area 3. The height of the tag is 1.1 m which has a 0.4 m height difference with fixed anchors.

### 3.2. Experimental Results for Identifying NLOS

In two experiments, use the abovementioned method to identify the NLOS for all anchors when a tag is moving along the reference path. The threshold used in those two experiments can be observed from a prior experiment. The tag moved at a uniform velocity in LOS environments (Area 1) in the prior experiment to measure a group of ranges. The variance of the group range data (0.004) is the threshold in this environment. The results are presented in the following two parts.

#### 3.2.1. Experimental 1

Figure 7 shows the result of NLOS identification for four anchors. We chose 5 sample points for each anchor, which contain LOS and NLOS. The Var($\Delta {\hat{d}}_{n}$) for sample points were shown in the figure and through the different values of Var($\Delta {\hat{d}}_{n}$) we can distinguish whether those points are in NLOS or not.

The accuracy of identification is calculated by dividing the total sample number by the number of misidentified cases, which is shown in the following Table 2:

#### 3.2.2. Experimental 2

Refer to Figure 7 and Figure 8, Table 2 and Table 3, that the method proposed in this paper is relatively accurate. The worst accuracy of the NLOS identification case is the Anchor 3 in both experiments, which is 90.18% and 88.65%, respectively. The accuracy of other anchors is all over 90% in two experiments.

The misidentified case includes cases in NLOS that did not recognise and in LOS but misjudged into NLOS, which named unidentified point and unexpected point in Figure 7 and Figure 8, respectively. The complex experiment environment containing computers, desks, chairs and other factors will increase the impact of multipath, which may lead to some unexpected points. The velocity of the tag is small during the turning, which is the possible reason for the unidentified point. Considering the impact of the factors above, this accuracy rate of NLOS identification is quite acceptable.

### 3.3. Mitigation Results for Delay Model (Experiment 1)

The experiment applies different position algorithms based on the number of NLOS in corresponding areas. According to the reference path and anchors’ position in experiment one, Area 1 has three anchors in LOS (Anchors 0, 1 and 2), which is enough to locate the tag accurately. For Area 2, the delay model was used to correct the range measured by Anchor 3. When the tag moves in Area 3, the UWB signal from Anchors 0 and 2 will be blocked by a 0.26 m concrete wall, the delay model is also applied to those measured data containing NLOS errors. In Area 4, using the range data from Anchors 0 and 1, which are in LOS, can compute two coordinates for the tag, then use the range data of Anchor 2 or Anchor 3 to select the tag’s actual coordinate. The worst scenario is in Area 5; the delay model must correct all four anchors in NLOS since a 0.16 m concrete wall blocked all the UWB signals. The following figure and table show the experiment results.

Figure 9 visually illustrates the effectiveness of the delay model proposed in this paper. The green line with the dot in Figure 9 shows the tag position and trajectory computed by the measured distance after the delay model correction. It explicitly shows that the position accuracy of the tag was improved in Areas 2, 3, and 5, which applies the delay model to correct data with NLOS delay. Table 4 is the detailly analysis of the experiment results. The first column of Area 1 is the position error using raw data from three LOS anchors (Anchors 0, 1 and 2) and one NLOS anchor (Anchor 3). The second column in Area 1 is the position error only using the range data from those three LOS anchors, and this group data can be used as the control group. It can be found that the NLOS error has a massive impact on position accuracy. With Anchor 3’s NLOS error in Area 1, the maximum position error is over 0.8 m, and RMSE is 0.555 m. The UWB IPSs perform better position accuracy after abandoning the range data from Anchor 3; the maximum error and RMSE reduce to 0.361 m and 0.106 m, respectively. The delay model minimises the affection of NLOS for areas that do not have enough LOS data to locate the tag. From Table 2, RMSE reduced from 0.364 m to 0.126 m in Area 2, from 0.853 m to 0.244 m in Area 3 and from 0.398 m to 0.276 m in Area 5. A similar phenomenon of error decreasing occurs in average and maximum error too. The position accuracy of UWB IPSs in NLOS after correction by the delay model is approaching the accuracy in LOS. Another phenomenon that can be found in Figure 9 is the location point of the tag in Area 3 dispersed more widely with the incident angle growth. So the Weight Least Square (WLS) was used in the next step to eliminate the effection of heteroscedasticity.

### 3.4. Mitigation Results for Delay Model with WLS (Experiment 1)

The hinge of WLS is to determine the exact weight for measured distance at different incident angles. The tag position kept jittering due to the measurement range affected by the noise, even though the tag was motionless. The jitter margin of range measurement positively correlates with the incident angle in NLOS caused by a wall. Thus, the variance of the range measurements in a position can be the weight’s reference.

In this experiment, weight for measured distance from the fixed Anchors 0, 1, 2 and 3 can be represented as:(25)$$W=diag\left[W_{0},W_{1},W_{2},W_{3}\right]$$

Setting the variance of measured distance in LOS as the criterion, the weight can compute by the following equation when Tag in NLOS with the incident angle θ.
(26)$$W_{\theta }=\frac{Var\left(d_{LOS}\right)}{Var\left(d_{\theta }\right)}$$
where $Var\left(d_{LOS}\right)$ is the variance of measured distance in LOS, $Var\left(d_{\theta }\right)$ is the variance of measured distance in NLOS with the incident angle θ, $W_{\theta }$ is the weight for this NLOS measurement range.

In order to find the relationship between weight and incident angle, 30 group data was gathered, which covered incident angles from 0° to 70°. Each group of data contains at least 60 samples, so $Var\left(d_{\theta }\right)$ can be computed for each angle group, and the weight for different incident angles can be drawn from Equation (26). The following Figure 10 illustrates the relationship between the incident angle and the weight. When the incident angle is less than 40°, the weight of the measured range can be set to 20; when the incident angle is greater than 40°, it is evident from the figure that the weight increased with the incident angle, and the weight can be calculated by the fitting equation.

The experiment in this part uses the same measured range data from Section 3.3. According to the relationships between θ and the $W_{\theta }$, assign corresponding weight to the measured range. The measured range with weight can calculate the tag’s position by Equation (17). The following Figure 11 and Table 5 show the position results of the delay model with the WLS algorithm.

The blue line with the dot in Figure 11 is the tag position calculated by the WLS algorithm. Compared with the green line (OLS algorithm), the position accuracy of WLS has significant improvement in Areas 2 and 3; also, the detailed data analysis in Table 5 can attest to that. In Area 3, the WLS significantly enhances the position precision, the maximum position error has a nearly 0.3 m reductions (0.603 m to 0.319 m), and the RMSE reduces from 0.244 m (OLS) to 0.132 m (WLS). The maximum position errors were almost equal in Areas 2 and 5, and the average errors and RMSEs slightly decreased using the WLS algorithm. Even the WLS algorithm did not contribute generously to the maximum error, but from average error and RMSE, the position accuracy was improved overall. Hence, the application of WLS further enhances the accuracy and precision of the UWB IPSs in NLOS environments.

### 3.5. Mitigation Results for Delay Model with WLS (Experiment 2)

To further verify the affection of the delay model and WLS algorithm, an improved experiment was designed in which the coordinate of Anchor 1 was changed to (0.49, 2.65, 1.5), resulting in only one LOS anchor in Area 3. That means the condition for UWB IPSs in Area 3 will be harsher than in Experiment 1. The following Figure 12 illustrates the coordinates of anchors (yellow square), reference path (purple dot line) and experiment results (red, green and blue line).

Referring to Figure 12, the red line represents the tag’s position and trajectory computed by OLS using raw range data which uses the same strategies in experiment 1. Green and blue lines represent the position and trajectory of the tag, which the delay model corrected the range data, but used OLS and WLS to compute the position, respectively. The main difference between this experiment and experiment 1 is in Area 3. By changing the coordinate of Anchor 1, the tag in Area 3 can only receive one LOS UWB signal. Even so, the delay model and the WLS can mitigate the NLOS error well, which can be observed from the contrast of the three trajectories. Detailed data can be found in Table 6. The second column data in Area 1 is data analysis for the tag’s position using only three LOS anchors’ range information, which can be the evaluation criteria for other groups. In Area 2, there is only Anchor 3 in NLOS, but this NLOS delay still leads to a maximum 0.416 m position error, the average error and RMSE all over 0.2 m. After using the delay model to correct the range data of Anchor 3, the maximum error reduces to 0.247 m; the average error and RMSE are around 0.1 m by the OLS algorithm. Changing the position algorithm to WLS can further improve the accuracy of the tag; the maximum error is only 0.195 m, and the average error and RMSE can reach the control group’s performance in Area 1.

In Area 3, despite moving Anchor 1 to the NLOS environment, the position accuracy remains largely untouched by this change compared to experiment 1. The reason may be that two walls block the direct path between the tag and Anchor 1 entirely, but by multipath, the tag and Anchor 1 still have regular communication. Accordingly, the measurement error of Anchor 1 should be stable, not leading to a vast negatively impact on position accuracy.

For other areas, the order of the performance of those position algorithms is delay model with WLS is the best, and then the delay model with OLS is better than OLS only. However, in Area 5, there is one exception, the maximum error of the delay model with WLS is 0.359 m which is bigger than 0.284 m for the delay model with OLS. The inappropriate weight of some points may cause this exception due to the complex indoor environments. In summary, the delay model with WLS algorithms can enhance the overall accuracy and precision of the UWB IPSs in the NLOS environment.

## 4. Discussion

Even though the performance of the delay model with the WLS algorithm is close to the LOS level, the RMSEs in Table 5 and Table 6 are all over 0.1 m in Areas 2, 3 and 5. Moreover, the jitter margin of the tag’s position in those NLOS areas is still more significant than that in the LOS environment. So some inherent defects have restricted the application of IPSs only using UWB technology. For instance, in the experiment environment, if the tag in Area 5 moves forward to the far right of the map, the communication between the tag and some anchors will be severed by more than one wall or some metals (refrigerator, metal cabinet, rebar in load-bearing wall). Therefore, the UWB IPSs can not provide the tag position on account of lacking enough measured range data in the hash indoor environment.

The same situation happened in Area 4; the signal from Anchors 2 and 3 will block by two walls. The delay model and WLS algorithms cannot handle the data like this. Therefore, in Area 4, we use a different strategy to calculate the tag position. In Area 4, the tag can receive two LOS signals from Anchors 0 and 1. These two measurement ranges in the LOS environments can obtain two coordinates. Range data from Anchor 2 or 3 can select the accurate coordinate for the tag from Anchor 2 or 3. The positioning result using this method in Area 4 can be observed in Figure 11 and Figure 12, and the accuracy was indeed improved than using raw data with OLS. Especially in Figure 12, the accuracy and precision of the tag’s position in Area 4 are all the better. This result was predictable because the correct range information calculated the tag’s position, so the position result should be as good as in Area 1. Only using range information from LOS anchors can improve the position accuracy if the system has more than one LOS anchor.

The geometric position of the anchors is another factor affecting the position accuracy of UWB IPSs [25]. In experiments 1 and 2, the tag, Anchors 0 and 2, were almost in a line when the tag was moving along the reference path in Area 2. According to the literature [26], these bad anchor placement positions will significantly reduce the position accuracy of the tag. Therefore, the tag in Area 2 can communicate with three LOS anchors, but the position accuracy is still not satisfying.

Even though the methodology used in this paper can improve the position accuracy and precision of UWB IPSs, it is impossible to settle all issues for indoor position systems only relying on one method or sensor. Future research will consider multisensor fusion to enhance position accuracy and system robustness. Different sensors have their own inherent strengths and weaknesses, and multi-sensor fusion allows for complementary advantages and system redundancy [27].

The IMU can autonomously provide the position information which is not disturbed by the outside environment. The IMU can provide acceleration, angle and altitude of the system at a high frequency, but the accuracy of the IMU can be seriously affected by cumulative accelerometer errors, gyroscope drift errors, magnetic field variations, etc. [28]. So IMU is a commendable sensor to support UWB IPSs in the NLOS environment but not for a long time. Another sensor we consider to support UWB IPSs is the wheel odometer. By the Differential Drive Kinematics model, two wheels odometer can offer the distance and yaw for the system. Wheeled odometers are similar to IMUs in that they provide accurate information to the system for a short period of time in the 2D plane, but without correction, the measurement error increases with time. Another problem with wheeled odometers is wheel slip, which can also affect positioning accuracy [29]. A further plan is to combine the camera with UWB/IMU/Odometer IPSs to enable mapping so that this system can operate well in harsh and unstructured environments. The fusion of these sensors can also overcome the problem of cameras failing to find feature points in single environments, leading to positioning failure [30].

## 5. Conclusions

This paper proposed a novel and simple method to identify and mitigate the NLOS for UWB IPSs. The strategy to correct the NLOS error is first to identify the NLOS delay by the sliding window, then update the measured range containing the NLOS error by the delay model, further using the WLS algorithm to calculate the tag’s position. The experiment results for accuracy of NLOS identification is approximately 90% in a harsh environment, mostly over 90% in a standard indoor environment. The performance of the NLOS delay model was impressive. According to the data in Table 4, Table 5 and Table 6, the accuracy of tag location in NLOS, which correction by the delay model, has been improved to a certain extent. Although the delay model revised the measured range and improved position accuracy, the location precision is still low. As the green line in Figure 10 and Figure 11, the large jitter margin of the position result remains a problem. The solution is by WLS algorithms to give different weights according to the UWB signal propagation incidence angle through the wall. The test results showed that the delay model with the WLS algorithm could improve the accuracy and precision of UWB IPSs close to the LOS level.

## Figures and Tables

**Figure 1 sensors-22-08247-f001:**
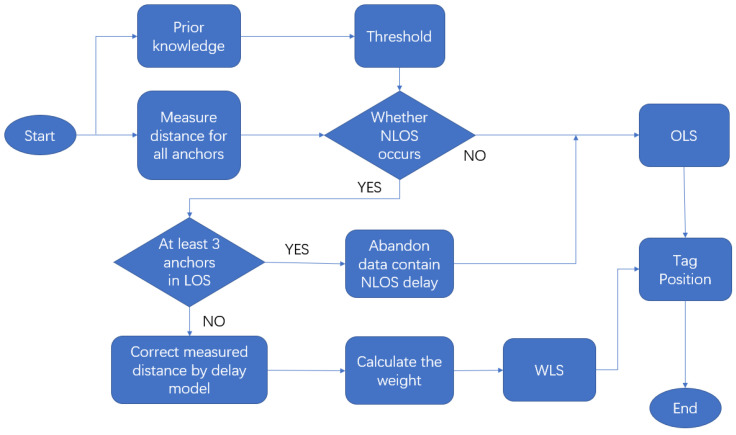
Flowchart for the proposed method.

**Figure 2 sensors-22-08247-f002:**
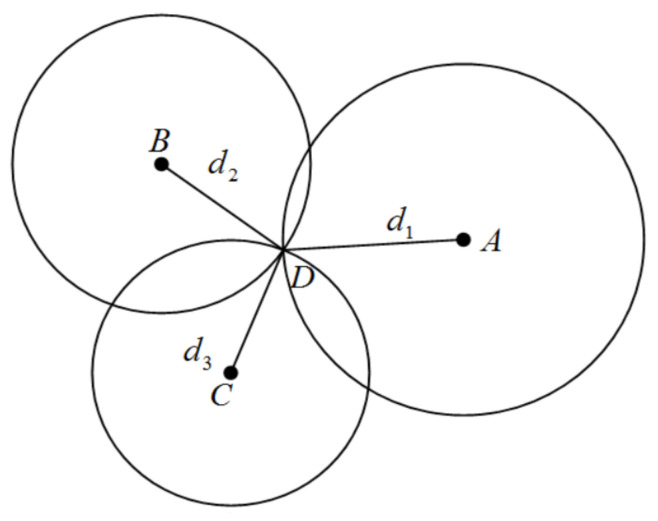
Geometric position of fixed anchors and the tag.

**Figure 3 sensors-22-08247-f003:**
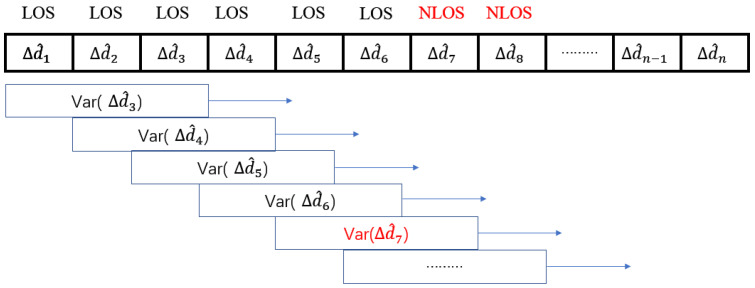
Sliding window method for NLOS identification.

**Figure 4 sensors-22-08247-f004:**
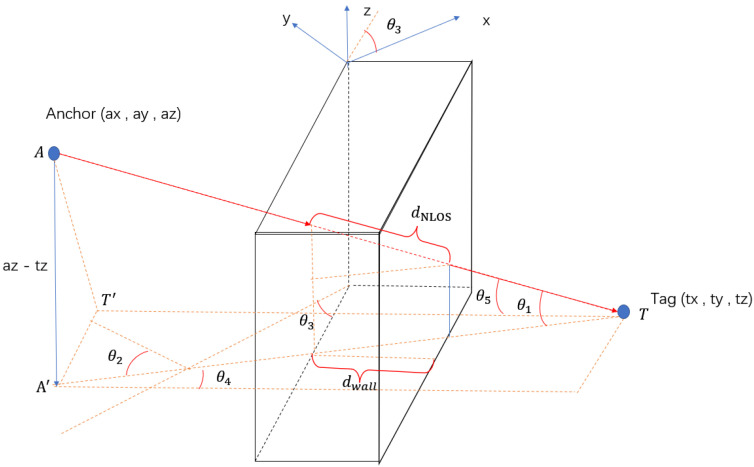
Delay model for a wall.

**Figure 5 sensors-22-08247-f005:**
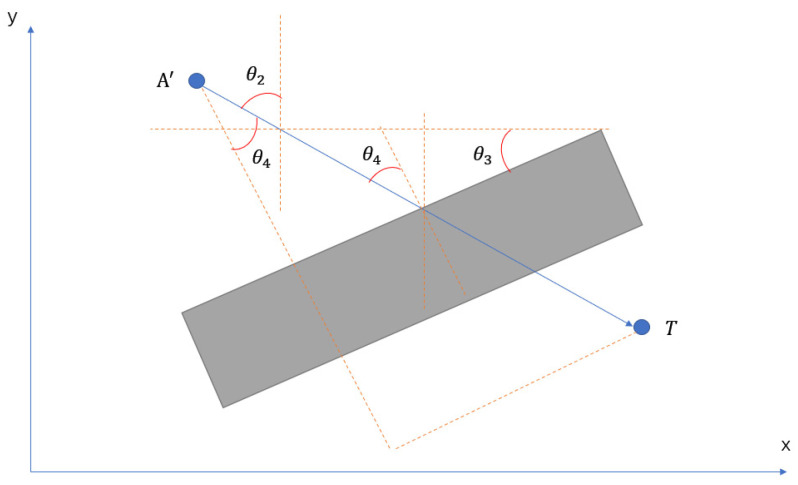
Projective image.

**Figure 6 sensors-22-08247-f006:**
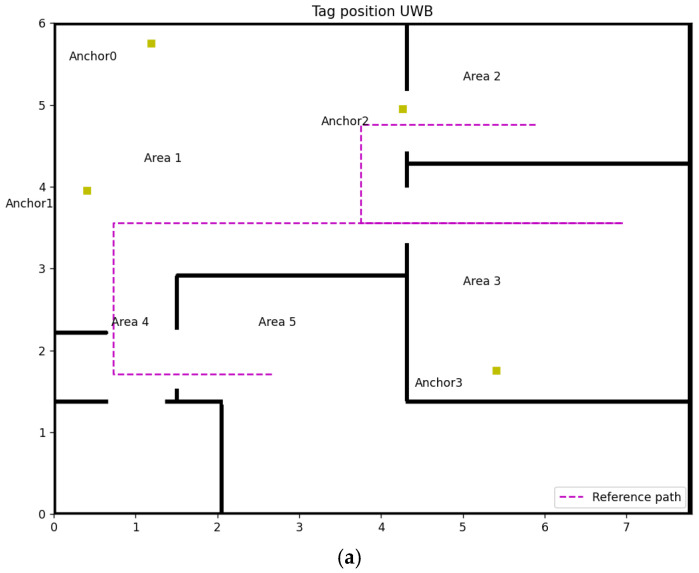
Experiment design: (**a**) Map of experiment 1, (**b**) Map of experiment 2, (**c**) Anchor 0 & 1, (**d**) Anchor 2, (**e**) Anchor 3, (**f**) Overall experiment environment.

**Figure 7 sensors-22-08247-f007:**
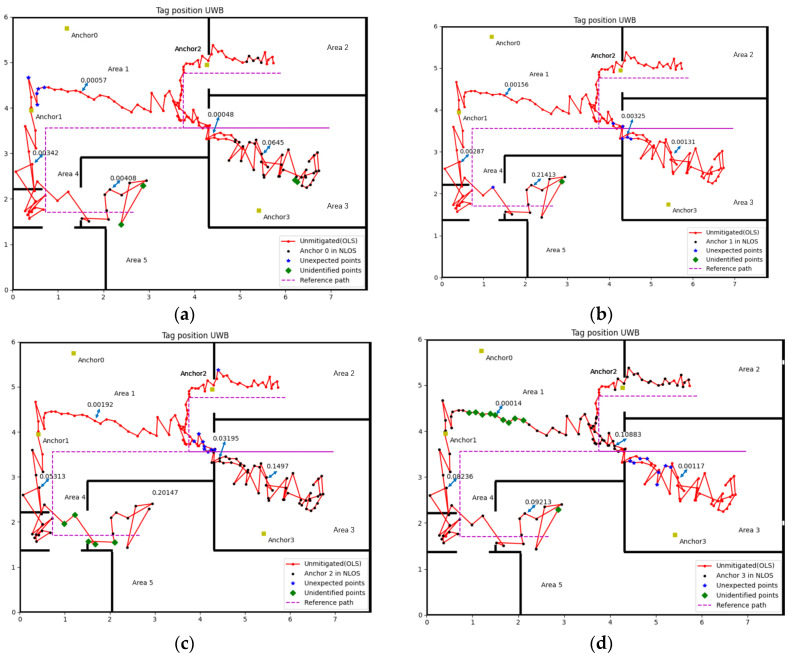
NLOS identification result (Experiment 1): (**a**) Anchor 0, (**b**) Anchor 1, (**c**) Anchor 2, (**d**) Anchor 3.

**Figure 8 sensors-22-08247-f008:**
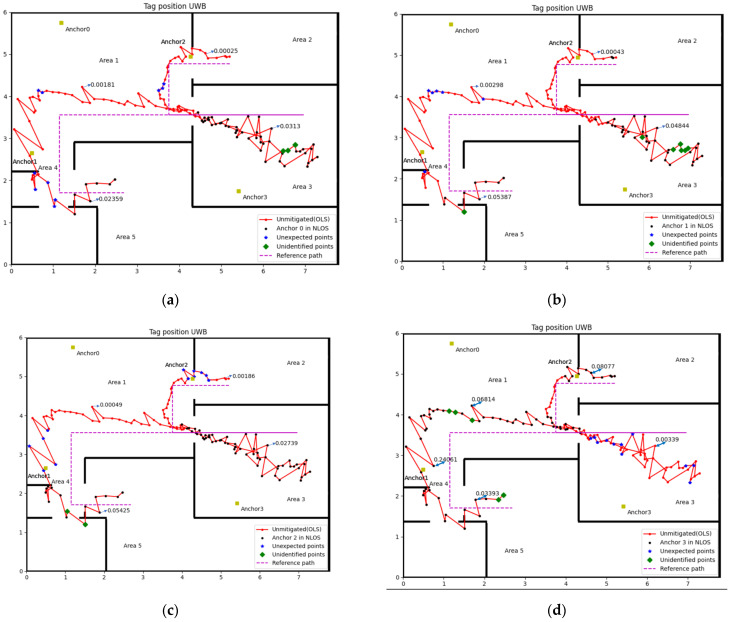
NLOS identification result (Experiment 2): (**a**) Anchor 0, (**b**) Anchor 1, (**c**) Anchor 2, (**d**) Anchor 3.

**Figure 9 sensors-22-08247-f009:**
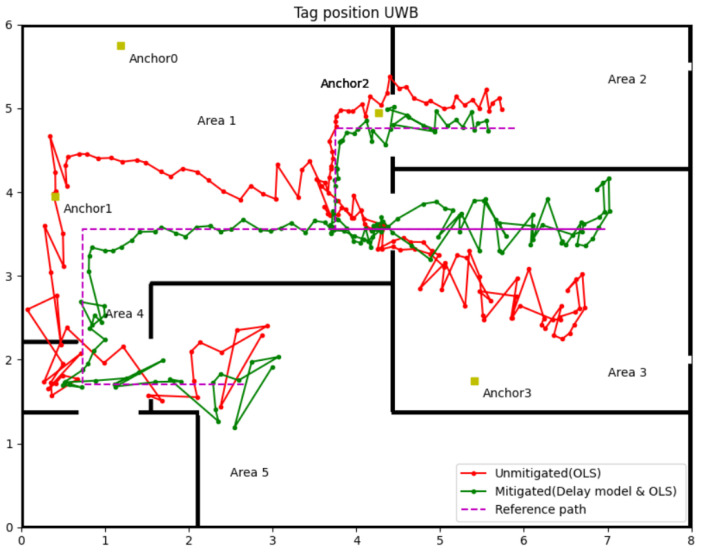
Tag position after mitigating by delay model with OLS.

**Figure 10 sensors-22-08247-f010:**
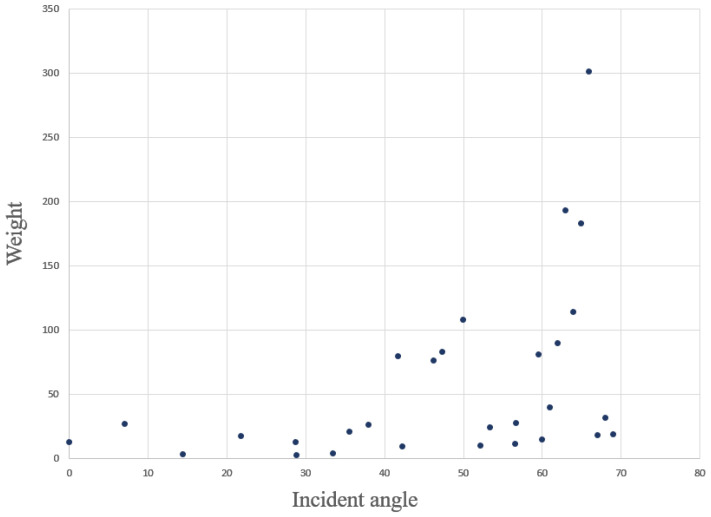
Relationship between *θ* and the $W_{\theta }$.

**Figure 11 sensors-22-08247-f011:**
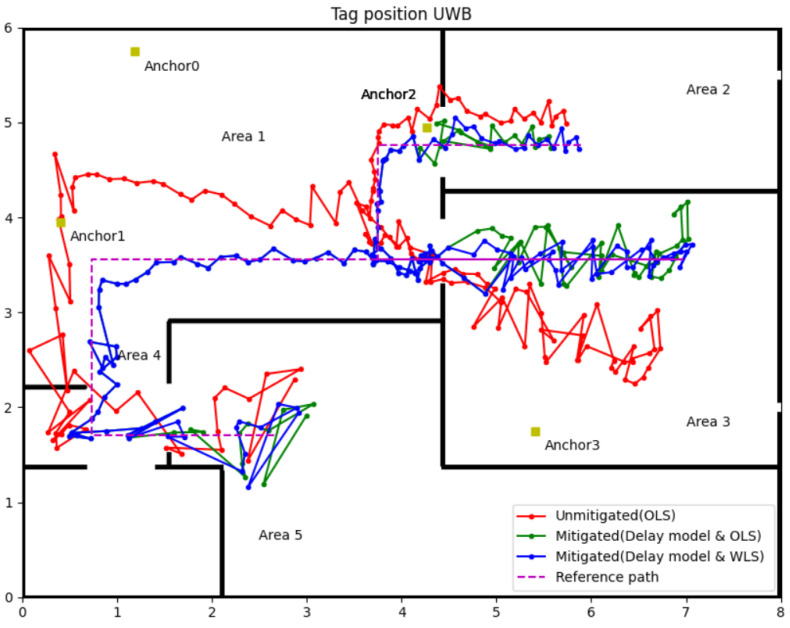
Experiment 1 results for the delay model with WLS.

**Figure 12 sensors-22-08247-f012:**
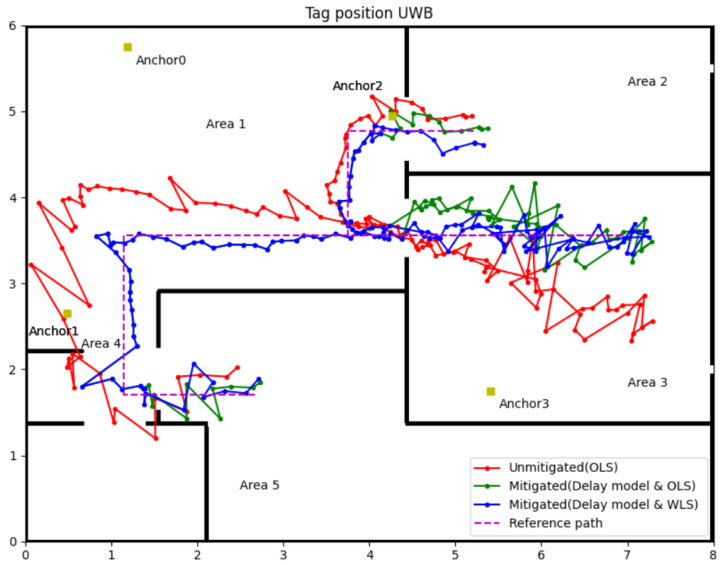
Experiment 2 results for the delay model with WLS.

**Table 1 sensors-22-08247-t001:** Anchors in NLOS in different Areas.

	Area 1	Area 2	Area 3	Area 4	Area 5
**Anchor** **(NLOS-Experiment 1)**	3	3	0 & 2	2 & 3	0, 1, 2 & 3
**Anchor** **(NLOS-Experiment 2)**	3	3	0, 1 & 2	2 & 3	0, 1, 2 & 3

**Table 2 sensors-22-08247-t002:** The accuracy rate of NLOS identification (Experiment 1).

Anchor	Anchor 0	Anchor 1	Anchor 2	Anchor 3
**Identification rate**	94.47%	95.71%	92.64%	90.18%

**Table 3 sensors-22-08247-t003:** The accuracy rate of NLOS identification (Experiment 2).

Anchor	Anchor 0	Anchor 1	Anchor 2	Anchor 3
**Identification rate**	90.71%	90.71%	90%	88.65%

**Table 4 sensors-22-08247-t004:** Position accuracy analysis for delay model.

Position Error	Area 1 (3 LOS Anchors)		Area 2 (3 LOS Anchors)		Area 3 (2 LOS Anchors)		Area 5 (0 LOS Anchors)	
	OLS	OLS (LOS Only)	OLS	Delay Model & OLS	OLS	Delay Model & OLS	OLS	Delay Model & OLS
**Max (m)**	0.849	0.361	0.616	0.257	1.313	0.603	0.692	0.521
**Min (m)**	0.048	0.004	0.212	0.003	0.153	0.016	0.033	0.013
**Average (m)**	0.499	0.075	0.349	0.094	0.786	0.201	0.339	0.213
**RMSE (m)**	0.555	0.106	0.364	0.126	0.853	0.244	0.398	0.276

**Table 5 sensors-22-08247-t005:** Position accuracy analysis for delay model with WLS (Experiment 1).

Position Error	Area 2 (3 LOS Anchors)		Area 3 (2 LOS Anchors)		Area 5 (0 LOS Anchors)	
	Delay Model & OLS	Delay Model & WLS	Delay Model & OLS	Delay Model & WLS	Delay Model & OLS	Delay Model & WLS
**Max (m)**	0.257	0.259	0.603	0.319	0.521	0.547
**Min (m)**	0.003	0.003	0.016	0.002	0.013	0.019
**Average (m)**	0.094	0.083	0.201	0.112	0.213	0.207
**RMSE (m)**	0.126	0.111	0.244	0.132	0.276	0.265

**Table 6 sensors-22-08247-t006:** Position accuracy analysis for delay model with WLS (Experiment 2).

Position Error	Area 1 (3 LOS Anchors)			Area 2 (3 LOS Anchors)		
	OLS (Raw Data)		OLS (LOS Only)	OLS	Delay Model & OLS	Delay Model & WLS
**Max (m)**	0.666		0.163	0.416	0.247	0.195
**Min (m)**	0.002		0.002	0.078	0.002	0.013
**Average (m)**	0.311		0.070	0.232	0.093	0.095
**RMSE (m)**	0.361		0.082	0.252	0.122	0.112
**Position Error**	**Area 3** **(1 LOS Anchors)**			**Area 5** **(0 LOS Anchors)**		
	**OLS**	**Delay Model &** **OLS**	**Delay Model &** **WLS**	**OLS**	**Delay Model &** **OLS**	**Delay Model &** **WLS**
**Max (m)**	1.231	0.603	0.362	0.511	0.284	0.359
**Min (m)**	0.034	0.009	0.001	0.043	0.004	0.010
**Average (m)**	0.522	0.237	0.105	0.240	0.134	0.118
**RMSE (m)**	0.638	0.276	0.130	0.272	0.163	0.159

## Data Availability

Not applicable.

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
