# Peer review of "A Succinct Method for Non-Line-of-Sight Mitigation for Ultra-Wideband Indoor Positioning System"

_sensors, 2022, doi:10.3390/s22218247_

Round 1
Reviewer 1 Report
Great job, congratulations. I suggest:
* improve the presentation of the image 6.c), that is, use a background that does not distract the reader's attention.
* In figures 7, 8, 9, 11 and 12 avoid that the labels are not crossed by the lines (anchor, area, etc.)
Author Response
Thank you for your comments.
In the revised version:
Figure 6 (c) was changed to use simple background.
Figure 7,8,9,11 and 12 was be modified to avoid the labels are crossed by the lines
Reviewer 2 Report
This paper proposed a method to identify NLOS and mitigate the error to improve UWB positioning performance. The discussion and proposed example demonstrate all the possible NLOS and LOS combinations, yet it is conducted over a small homogeneous area.
The paper introduces the problem well, one missing element from the discussion are the IMU, SLAM and map matching techniques which are used for indoor positioning. Those are mature, widely used and should be discussed.
It would be useful to describe novelty/need better, for example, what indoor market segment (commercial application) is this research focusing on.
The algorithm is clearly described, and Fig 3 would benefit from additional information in section 3, complementing figures 7+ as \epsilon_{NLOS} threshold and values are not clear. This could be presented as additional plot of \epsilon_{NLOS} vs time or \epsilon_{NLOS} vs d\delta d_n
Importantly paper discusses the means of correcting the range and providing the corrected range. This approach has two limitations:
* A specific material and thickness were assumed for the delay model. How does it work in different or varied environments?
* It assumes that NLOS is a direct but delayed signal. This is possible, given UWB wideband signal characteristics but not guaranteed - materials such as metal will reflect the signal so it is possible to receive bounced signal. This is briefly discussed in l 405 but not addressed.
I would suggest at least expanding the Algorithm section to discuss those limitations.
Few more comments below:
l 44 The IMU, SLAM and map matching approaches are missing. Those are independent techniques and not only auxiliary as indicated in l 79.
l xx How detection threshold is estimated?
l 275 figure 7 is a bit unclear. It would be useful to have Var(d) overlayed on the patch to understand how the filter presented in fig 3 performs
l 280 how the identification rate was calculated.
l 356 Fig 10, I see no relationship between angle and weight (R2 would be close to 0). The results indicate that weighting is effective so maybe a different plot or better explanation is needed.
Author Response
- At the end of the algorithm section (line 254), a section has been added to explain the scenarios in which the model is applicable and the drawbacks of the model
- IMU and SLAM are well-established techniques for indoor positioning, but as these are not used in this paper, we will consider the fusion of these sensors in future research. Therefore, we have added information on IMU, camera and wheeled odometer to the discussion section. (line 470 – 487)
- Added explanation of how the threshold is measured in lines 281 to 284
- For Figures 7 and 8, a few different cases of sampling points have been selected, and the Var() of these points have been labelled in the figures, while an explanation of this section has been added in lines 289 to 292
- Lines 293 and 294 explain how identification accuracy is calculated
- Added lines 371 to 374 to explain the relationship between UWB incidence angles and weights in experimental operations
- Please see the resubmitted version and the cover letter for a more detailed explanation for comment 4.
Reviewer 3 Report
The authors present a NLOS detection mechanism to improve the localization performance of UWB systems in indoor locations. The paper is technically sound and relatively well presented. The NLOS mechanism uses detection of the variance of second order statistics over a slicing window. The method produces more accurate localization in NLOS environments. However, the experimental results show still large experimental values of the positioning error. The value of the paper lies more on the experimental results than on the theoretical aspects that look a bit basic.
Author Response
Thank you for your comments, and I will try to improve the position accuracy in the future research